# Effects of the Whole-Body Vibration Exercise on Sleep Disorders, Body Temperature, Body Composition, Tone, and Clinical Parameters in a Child with Down Syndrome Who Underwent Total Atrioventricular Septal Defect Surgery: A Case-Report

**DOI:** 10.3390/children10020213

**Published:** 2023-01-25

**Authors:** Luiza Torres-Nunes, Patrícia Prado da Costa-Borges, Laisa Liane Paineiras-Domingos, José Alexandre Bachur, Ana Carolina Coelho-Oliveira, Danúbia da Cunha de Sá-Caputo, Mario Bernardo-Filho

**Affiliations:** 1Laboratory of Mechanical Vibrations and Integrative Practices, Department of Biophysics and Biometrics, Roberto Alcântara Gomes Institute of Biology and Piquet Carneiro University Polyclinic, State University of Rio de Janeiro, Rio de Janeiro 20950-003, Brazil; 2Program of Postgraduate Degree in Clinical and Experimental Pathophysiology, State University of Rio de Janeiro, Rio de Janeiro 20550-170, Brazil; 3Department of Physiotherapy, Multidisciplinary Institute of Rehabilitation and Health, Federal University of Bahia, Salvador 40110-909, Brazil; 4Department of Physiotherapy, University of Franca, Franca 14404-600, Brazil

**Keywords:** down syndrome, child, congenital heart, physical exercise, whole-body vibration

## Abstract

Background: The health and developmental issues of people with Down syndrome (DS) are complex and are associated with many medical, psychological, and social problems from childhood through into adulthood. DS children have an increased risk of multiorgan comorbidities, including congenital heart disease. Atrioventricular septal defect (AVSD) is a congenital heart malformation that often occurs in DS people. Aim: Physical activity and exercise are recommended for patients with cardiovascular disease and are considered to be the gold standard of cardiac rehabilitation. Whole-body vibration exercise (WBVE) is considered a form of exercises. The aim of this case report is to show the effects of WBVE on sleep disturbances, body temperature, body composition, tone, and clinical parameters in a child with DS with corrected total AVSD. The subject is a 10-year-old girl, with free-type DS, who underwent surgery to correct a total AVSD at 6 months. She underwent periodic cardiological monitoring and was released to perform any type of physical exercise, including WBVE. WBVE improved sleep quality and body composition. Conclusion: WBVE leads to physiological effects that benefit the DS child.

## 1. Introduction

Down syndrome (DS) is a genetic condition caused by the tripling of chromosome 21 (T21) and is characterized by a wide range of physical and neurodevelopmental disabilities [1]. Researchers are interested in DS since it is the most frequent genetic mutation [2].

DS children are more likely to develop multiorgan comorbidities [3]. Congenital cardiac disease is frequent in DS children, with ventricular septal defects being the most prevalent type [4]. A common cardiac deformity in DS patients, atrioventricular septal defect (AVSD), accounts for up to 5% of all congenital heart problems [5]. Total AVSD, the most common form in the DS group, has a worse prognosis than partial AVSD [5]. Infant mortality before surgery is high despite the postoperative mortality being low [5]. For the treatment of managing and lowering morbidity and mortality in DS children, early diagnosis of congenital heart disease and other comorbidities is advised [3,4]. DS children also have a higher probability of epilepsy, autoimmune diseases, celiac disease, arthropathy, thyroid dysfunction, diabetes mellitus, dermatological changes, auditory and visual abnormalities, and autistic spectrum disorder [3].

All patients with cardiovascular diseases are recommended to engage in controlled physical activity and exercise, which are regarded as the gold standard of cardiac rehabilitation [6] and offers a direct impact on morbidity and mortality, as well as in improving quality of life [7]. Exercises such as whole-body vibration exercise (WBVE), a type of moderate-intensity physical activity, can improve health [8,9]. Among the main effects of the WBVE are increased peripheral blood circulation [10,11] and lymph [12], decrease of pain [13,14], spasticity [15], positive effects on the neuromuscular system [16], and improvements of cognitive functions. [17].

WBVE can be a helpful alternative for monotonous activities performed during cardiac rehabilitation [18]. WBVE occurs when the individual is exposed to mechanical vibrations generated in a vibrating platform at a specific frequency and amplitude, adapted to each individual and their clinical condition [19] during the systemic vibratory therapy [20,21]. WBVE compared to no training [22,23,24] or as an adjunct to exercise, [25,26] has shown positive short-term effects on static balance within children and adolescents with DS, [22] on lower limb muscle strength in DS children [25,26], and subtotal bone mineral density and subtotal bone mineral content in DS adolescents [24]. There have been no short-term effects due to the WBVE that been represented in the body fat of DS children [26] or the body fat and lean body mass in DS adolescents [23]. 

Eid et al. [25] investigated if the WBVE could improve standing balance and muscle strength in 30 DS children (8–10 years old). The control group received a designed physical therapy program, whereas the study group received the same program given to the control group, in addition to WBVE. Both groups received the treatment sessions three times per week for six successive months. Significant improvements in stability indices and muscle strength after treatment were demonstrated in each group, with significantly greater improvements seen in the study group when compared with the control group (*p* < 0.05), suggesting that WBVE may be a useful intervention modality to improve balance and muscle strength in DS children.

Considering an increased prevalence of obesity and muscle weakness in DS children, Emara et al. [26] examined the effects of a 12-week WBVE on the body composition and muscle strength of 31 DS children (10–12 years old) divided randomly into 2 groups, regarding the higher prevalence of obesity and muscle weakness in DS children. The trial group received WBVE in addition to the physical therapy program that was offered to the control group. For three months in a row, there were three treatment sessions every week. The results revealed that there was a significant increase in lower limb strength in both groups (pre and post treatment). Comparison concerning post treatment results revealed more increase in the lower limb strength in the WBVE group, suggesting that this modality may improve lower extremities muscular strength in obese DS children. 

Finally, a study proposed by Eid et al. [27] investigated the effects of isokinetic training on muscle strength and postural balance in 31 DS children (9 to 12 years old), divided randomly into 2 groups. The study group had isokinetic training 3 days per week for a period of 12 weeks, in addition to the same physical treatment as the control group underwent. Both groups had significant improvements in postural balance and the peak torque of their knee flexors and extensors (*p* < 0.05), with the study group experiencing significantly greater improvements than the control group (*p* < 0.05). These outcomes indicated that participation in the isokinetic training program induced greater improvements in muscle strength and postural balance in DS children. Therefore, the WBVE can be suggested for this population as an effective training modality. 

## 2. Case Report

### 2.1. Ethical Issues

This study was approved by the Ethics Committee of the *Pedro Ernesto University Hospital*, registered under number 30649620.1.0000.5259. 

The legal guardian read and signed a Free and Informed Assent (FIA) before any procedure was performed. All procedures performed were considered non-invasive and demonstrated no obvious risk.

The data collected, as well as the interventions, were carried out at the Laboratório de Vibrações Mecânicas e Práticas Integrativas (LAVIMPI), Policlínica Universitária Piquet Carneiro, Universidade do Estado do Rio de Janeiro, Brazil.

### 2.2. Identification and Clinical Conditions 

Female patient, 10 years old, with free-type DS. The diagnosis of DS occurred in prenatal exams during pregnancy, which also identified total AVSD. At 6 months of age, she underwent surgery to correct the total AVSD and since then she has been undergoing periodic cardiological follow-up and is stable without medication. She was cleared by the cardiologist to perform any type of physical exercise, including WBVE.

### 2.3. Parameters Evaluated and Instruments Used 

#### 2.3.1. Initial Assessment

Before performing the first intervention, the legal guardian signed the FIA authorizing the participation, and then an anamnesis was carried out to collect data and information from pregnancy, birth, development, and physical condition until the moment of the initial assessment.

#### 2.3.2. Body Temperature

Body temperature (BT) was measured by infrared thermography (IT) of the anterior region of the neck [28], before and 15 min after the intervention. IT was performed with a FLIR Systems camera, E40, Wilsonville, OR, USA, in a windowless room with air conditioning at a constant temperature of 23 °C, with 45 to 55% humidity, and the captured images were processed using the FLIR ResearchIR Max software (version 4.40.4.17, Sweden) [29]. The camera was placed at 1 m between the camera and the cricoid cartilage of the patient. To perform the IT, the legal guardian was instructed to remove the blouse of the child and any necklaces, so that they would not interfere with the image capture. The patient was positioned in a sitting position with support in the head region so that she could maintain a satisfactory cervical extension for capturing the images [30]. 

#### 2.3.3. Sleep Disorder

To evaluate the sleep disorder, the Reimão and Lefevre Children’s Sleep Questionnaire (QRL) [31] and the Children’s Sleep Disorders Scale (SDSC) [32] were applied. Both QRL and SDSC were applied before the first intervention and after the second intervention.

#### 2.3.4. Body Composition

Body composition (BC) analysis was performed before the first intervention and after the second intervention, using the bioelectrical impedance In Body 370, Korea [33].

#### 2.3.5. Tone Analysis

The tonus was assessed before and after the first intervention and after the second intervention, through muscle palpation and passive joint movements [34,35].

#### 2.3.6. Clinical Parameters

Clinical parameters such as blood pressure (BP), heart rate (HR), and respiratory rate (RR) were evaluated 3 times before and 3 times after the interventions and the averages were recorded. Oxyhemoglobin saturation (SatO_2_) was checked once before and once after the intervention. BP and HR were measured on the right arm (automatic BP monitor, OMRON, model HEM7113, Brazil) [36]. During the interval of BP measurements the children remained seated and the number of respiratory incursions performed was recorded for 1 min [37]. To measure SatO_2_, the child remained seated and the finger oximeter (Medical Rossmax model SB100, Taiwan) was placed on the index finger [38]. 

### 2.4. WBVE Intervention Protocol

The child performed two interventions of WBVE on a side alternating vibrating platform (Novaplate Fitness Evolution^®^, DF Produtos Hospitalares Ltd.a, São Paulo, Brazil), with a frequency of 5 Hz, peak-to-peak displacement of 2.5 mm, five sets of 30 s with vibration, and 1 min of rest were performed. The first session (WBVE 1) in the sitting position on the ancillary chair with feet on the base of the vibrating platform and the second session (WBVE 2) sitting directly on the base of the platform. The interval between the WBVE 1 and WBVE 2 was 15 days. 

## 3. Results

For the analysis of quantitative data, the mean and standard deviation of the collected data were used, and for the analysis of qualitative data, a comparison was made between the collected data. 

### 3.1. Clinical Parameters

Data collected before WBVE 1 were to BP, 104 × 63, 108 × 71, 114 × 68 mmHg; to HR, 79, 93, 74 bpm; RR 15, 15, 14 ipm; and to SatO_2_, 99%. After the WBVE 1 were to BP, 114 × 60, 101 × 62, 122 × 54 mmHg; to HR, 80, 77, 71 bpm; RR 16, 14, 15 ipm; and to SatO_2_, 99%. The mean and standard deviation of these findings are shown in Table 1.

Before the WBVE 2, the BP was 110 × 64, 109 × 51, 111 × 56 mmHg; HR 74, 78, 74 bpm; RR was 15, 15, 14 ipm; and SatO_2_ was 99%. After the WBVE 2, the BP was 107 × 57, 134 × 86, 114 × 56 mmHg; HR 73, 51, 77 bpm; the RR 15, 15, 15 ipm; and SatO_2_ was 99%. The mean and standard deviation of these findings are shown in Table 2.

When comparing the data, it is possible to qualitatively confirm that systolic pressure increased post WBVE 1 compared to pre WBVE 1, and that systolic and diastolic pressure increased post WBVE 2 compared to pre WBVE 2. This shows that, as would be expected after performing physical exercise, WBVE had a clinical impact on cardiovascular function. However, the changed parameters did not approach alarming levels.

There was a reduction in HR post WBVE 1 compared to pre WBVE 1, as well as in the HR post WBVE 2 compared to pre WBVE 2. The RR remained the same in pre WBVE 1 and post WBVE 1, as well as in pre WBVE 2 and post WBVE 2 and Sat0_2_ remained the same in all collections.

### 3.2. Infrared Thermography

Before the first IT analysis, the anterior region of the neck showed a mean temperature of 34 °C, with a center temperature of 34.5 °C, a maximum temperature of 35.6 °C, and a minimum temperature of 30 °C. After WBVE 1, the IT analysis of the anterior region of the neck showed a mean temperature of 33.9 °C, a center temperature of 34.2 °C, a maximum temperature of 35.9 °C, and a minimum temperature of 31.3 °C. Before the WBVE 2, the IT analysis showed a mean temperature of 34.4 °C, a center temperature of 34.0 °C, a maximum temperature of 36.0 °C, and a minimum temperature of 32.3 °C. After WBVE 2, the IT analysis of the anterior region of the neck showed a mean temperature of 34.8 °C, a center temperature of 34.3 °C, a maximum temperature of 36.1 °C, and a minimum temperature of 33.6 °C. Figure 1 shows the data obtained with IF.

It is possible to verify that the maximum and minimum temperatures, as well as the mean temperature, were slightly increased post WBVE 1 compared to pre WBVE 1. Pre and post the WBVE 2, the center temperature remained stable.

### 3.3. Tone

Before WBVE 1, palpation of the muscles showed bilateral hypotonia of the deltoid, bilateral biceps brachii (BB), bilateral triceps brachii (TB), bilateral abdominals, bilateral hamstrings, and left triceps surae (TS); as well as normal tone of the bilateral pectorals, bilateral abdominal obliques (BO), bilateral quadriceps, and right TS. In passive joint movements, the girl presented hypotonia of bilateral shoulder flexors (SF) and shoulder extensors (SE), bilateral wrist extensors (WE), bilateral hip flexors (HF), bilateral hip adductors (HA), bilateral knee flexors (KF), bilateral ankle dorsiflexors (AD), and bilateral ankle plantar flexors (AP); as well as normal tone of bilateral elbow flexors (EF), bilateral elbow extensors (EE), and wrist flexors (WF).

After WBVE 1, the palpation of the muscles showed hypotonia in the bilateral deltoid muscles, bilateral BB, bilateral TB, and left TS; as well as normal tone of the other groups. In the joints, he presented hypotonia in the bilateral SF, bilateral SE, bilateral WE, bilateral HA, bilateral KF, bilateral AD, and bilateral AP; as well as normal tone of bilateral EF, bilateral EE, bilateral WF, and bilateral HF.

After WBVE 2, muscle tone was hypotonic in the bilateral deltoid, bilateral BB, bilateral TB, bilateral pectoral, bilateral TS, and bilateral hamstrings; as well as normal tone in the abdominals, bilateral obliques, and bilateral quadriceps. Joint tone was hypotonic in the bilateral SF, bilateral SE, bilateral EF, bilateral WE, bilateral HF, bilateral HA, bilateral KF, left AD, and left AP; as well as normal in the bilateral EE, bilateral WF, right AD, and right AP. Table 3 shows the tonus data collected in muscle palpation and Table 4 shows the tonus data collected through passive joint movement.

After WBVE 1 and WBVE 2, it is possible to verify that there were slight alterations in the muscle tone. The changes in tone observed on the palpation of the muscle belly included: bilateral pectoral and abdominal from normal to hypotonic after the second intervention; bilateral hamstring from hypotonic to normal after WBVE 1; and right triceps surae from normal to hypotonic after WBVE 2. As for the tone verified through passive joint movement, it is possible to verify the changes: flexion of the left elbow from normal to hypotonic after WBVE 2; bilateral elbow extension from normal to hypotonic after WBVE 2; bilateral hip flexion from hypotonic to normal after the first intervention; and change in dorsiflexion and plantarflexion of the right ankle from hypotonic to normal after WBVE 2.

### 3.4. WBVE Results in Body Composition

Before WBVE 1, the body composition (BC) was 118.5 cm of height, 26 kg of body mass, 18.1 kg of lean mass, 19.1 kg of fat-free mass, 1.02 kg of bone mass, 14.2 kg of water total body, 3.7 kg of protein (normal), 1.23 kg of minerals (normal), 6.9 kg of body fat (excessive), 9.2 kg of muscle mass, body mass index of 18.5 kg/m2 (normal), body fat percentage of 26.5% (much higher), and a waist-to-hip ratio 0.76 (normal). After WBVE 2, the BC showed 119 cm of height, 25 kg of body mass, 18.4 kg of lean mass, 19.4 kg of fat-free mass, 1.03 kg of bone mass, 14.4 kg of body water total, 3.8 kg of protein (normal), 1.23 kg of minerals (normal), 5.6 kg of body fat (normal), 9.4 kg of muscle mass, body mass index of 17.6 kg /m2 (normal), body fat percentage of 22.2% (above), and a waist-to-hip ratio 0.76 (normal) (Table 5).

Height, lean mass, fat-free mass, bone mass, total body water, proteins, and muscle mass all increased after WBVE, according to data collected pre WBVE 1 and post WBVE 2. Post WBVE 2, body weight, body fat, body mass index, and fat percentage all decreased. Minerals and the waist-hip ratio remained stable.

### 3.5. WBVE Results in Sleep Disorder

#### 3.5.1. QRL

Post the WBVE 2 session, a new round of QRL responses was collected. Only five questions changed, suggesting that the sleep pattern had improved. After two sessions of WBVE, the child under investigation stopped urinating during naps, woke up in the morning in less than 15 min, did not arrive late to school from oversleeping despite family members waking up, did not have any trouble staying awake at school in the mornings, and had no need or desire to sleep longer.

#### 3.5.2. DSDC

In the DSDC performed before WBVE 1, the total score (TS) was 45; sleep initiation and maintenance disorder (SIMD) 15; sleep breathing disorders (SBD) 5; arousal disorders (AD) 6; sleep-wake transition disorders (SWTD) 6; excessive daytime sleepiness (EDS) 11; sleep hyperhidrosis (SH) 2. In the DSDC performed after WBVE 2, the TS was 43; SIMD 14; SBD 5; AD 3; SWTD 10; EDS 9; SH 2. These findings are also in Table 6.

Comparing the results before WBVE 1 and after WBVE 2, it is possible to verify that, qualitatively, there was a decrease in TS, SIMD, AD, and EDS as well as an increase in SWTD after WBVE 2. The SBD and SH did not change.

## 4. Discussion

The current case report shows that the WBVE can be used in DS children who have undergone surgery to correct a total AVSD. In the current case report, even for the child performing only two interventions, the WBVE provided benefits and positive outcomes of quick and easy execution.

WBVE is a potential intervention for improving motor function and bone growth in children with disabilities, although most of the evidence comes from adult trials. It is crucial to understand the mechanisms of both disabled and non-disabled children. WBVE appears to be a safe intervention and the protocol should be appropriate for both disabled and non-disabled children, considering the biomechanical parameters of each patient [39]. The outcomes of the current study support the findings of Saquetto et al. [40], that in a systematic review it is suggested that WBVE has positive effects on body composition, bone mineral density, and balance in DS children and DS adolescents. Additionally, Gusso et al. [41] performed vibratory training in 14 DS adolescents. On the base of the platform, the teenagers stood with their feet apart and their knees slightly bent. Each participant began their training with three series of 1-min sessions at a frequency of 12 Hz. The duration and intensity were gradually lengthened. For 20 weeks, there were 4 sessions every week. After the fourth week, all participants completed 3 sets of 3 min each at 20 Hz, 1 mm of amplitude, and 3 min of rest in between each set. A greater distance was traveled throughout the six minute walk test as a result, and muscle strength also increased.

Partial AVSD is more common in non-chromosomal cases, while total AVSD is more common in DS people [42]. The AVSD is the most prevalent type of congenital heart malformation lesion [43]. All patients with cardiovascular disorders should engage in appropriate and controlled physical activity and exercise, which is considered the gold standard of cardiac rehabilitation [6] and has a considerable impact on quality of life, morbidity, and mortality [17].

Thermoregulatory mechanisms are responsible for maintaining the optimal temperature of the body. Exercise speeds up oxygen transport and blood circulation, which raises metabolism and, ultimately, body temperature [44]. Dębiec-Bąk et al. [28], examined 36 people of both sexes, 18 DS individuals and 18 individuals without DS. All performed 45 min of identical training, with the necessary adaptations, twice a week for a period of 30 days. Before exercise, the temperature in all body regions was lower in DS individuals. After 5 min of training, the temperature decreased in all parts of the body in both groups, with the greatest decrease in the DS group. However, 15 min after training, the group without DS had body temperatures similar to the initial measurement, while the DS group temperatures did not return to baseline values. The findings presented in Figure 1 agree with the results reported by Dębiec-Bąk et al. [28].

Hypotonia is the primary cause of balance and postural control disorders in DS children and adolescents [45]. Skeletal disorders such as hip and patellar dislocation, as well as flat feet, are also common [46]. Hernandez-Reif et al. [47], conducted a study in DS children to evaluate the effect of massage therapy on motricity and muscle tone. When compared to the control group, the massage therapy group showed greater gains in fine and gross motor function, as well as alterations in hypotonicity. The findings suggest that including massage therapy in an early intervention program can improve motor functioning and modulate muscle tone in DS children. Song et al. [48] evaluated the effects of WBVE on trunk and lower limb muscle tone and activation, balance, and gait in a child with spastic diplegia cerebral palsy. WBVE decreased muscular tone in the hip extensor and plantar flexor of the right ankle; increased muscle activation in the bilateral erector spinae, rectus abdominis, rectus femoris, tibialis anterior, and gastrocnemius; and improved gait speed and balance. As a result, the authors concluded that WBVE can be used to safely and effectively maintain and increase physical performance and can be included in rehabilitation programs. Small changes in muscular tone were seen in this case study. However, considering the evidence assessed, it was agreed that the modulation of the aspect of hypotonia in DS could not be determined in only two WBVE sessions. 

Weight gain and growth are measures of overall health in DS people. Obesity affects 25% of DS children and 50% of DS adults, exacerbating a range of issues such as obstructive sleep apnea, diabetes, and cardiopulmonary conditions. Obesity diagnosis and early intervention to encourage healthy diet and exercise routines are essential for managing weight gains [46]. González-Agüero et al. [23], investigated how the 20-week WBVE affected the body composition of DS young people. Thirty DS adolescents were allocated into the control and WBVE groups. The WBVE group showed a greater reduction in body fat in the upper limbs and a tendency to increase the percentage of total lean body mass, which may be useful to improve body composition in this population. The increase in height, lean mass, fat-free mass and bone mass, proteins, total body water and muscle mass was observed in this case study. The decrease in body mass was also identified. This is consistent with research findings that suggest a positive effect of systemic vibratory therapy on the BC of DS children.

Sleep difficulties are another prevalent issue among DS children. Choi et al. [49] conducted a study in which 88 caregivers of Korean DS children aged 5 to 17 years old completed the Children’s Sleep Habits Questionnaire (CSHQ) to analyze the sleep characteristics of these children. The result showed that 83% of children with DS had sleep disorders. In this case study, an improvement in sleep initiation and maintenance, sleep agitation, and excessive sleep during the day was observed after the proposed protocol. There is presently no data in the literature describing WBVE protocols developed for DS children and analyzing sleep disorders.

The limitations of the present study are related to the quantity of interventions and any potential bias in the method used to measure muscle tone. Additionally, if other biomechanical factors and various WBVE protocols had been considered, the outcomes might have been different.

Another limitation of this case report that should be highlighted is the assessment of cognitive function. Although there is an understanding of the importance of identifying the cognitive potential of DS children, this ability was not considered as a criterion for participation in this protocol, because this evaluation and consideration of its consequences in the development of DS children requires qualified professionals.

The strength of this case study is that this is the first one to consider the effect of WBVE on a DS child who underwent total AVSD correction regarding CP, BT, muscle tone, BC, and sleep quality. This case report helps to fill part of the gap in the research field related to the effects of WBVE on CP, BT, tone, BC and sleep quality in DS children.

## 5. Conclusions

Considering the effects of the WBVE, it is possible to conclude that this type of exercise would promote biological effects with relevant physiological responses in blood pressure, which is an expected effect in the practice of physical exercise. The center temperature, mean, maximum and minimum temperatures verified 15 min after the interventions were also higher, confirming DS-related thermal mis adjustment. In addition, WBVE promoted a small modulation in tone and altered various parameters related to the BC. Moreover, there was also an improvement in the sleep pattern.

## Figures and Tables

**Figure 1 children-10-00213-f001:**
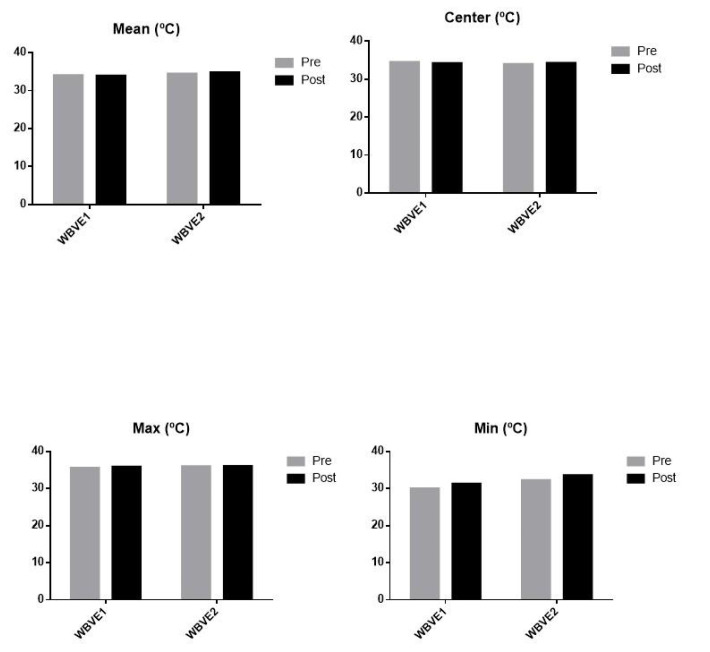
WBVE results of the anterior region of the neck assessed by infrared thermography. WBVE 1 = whole-body vibration exercise sitting on the auxiliary chair; WBVE 2 = whole-body vibration exercise sitting directly on the base of the platform; Max = maximum temperature; Min = minimum temperature.

**Table 1 children-10-00213-t001:** Results of the first WBVE intervention on clinical parameters.

	Pre WBVE 1		Pos WBVE 1	
BP (mmHg)	Systolic	Diastolic	Systolic	Diastolic
	108 ± 5.03	67 ± 4.04	112 ± 10.59	58 ± 4.16
HR (bpm)	82 ± 9.84		76 ± 4.78	
RR (ipm)	14 ± 0.57		14 ± 1.15	

WBVE 1 = the first intervention of whole-body vibration exercise; BP = blood pressure; HR = heart rate; RR = respiratory rate; bpm = beats per minute; ipm = incursions per minute.

**Table 2 children-10-00213-t002:** Results of the second WBVE intervention on clinical parameters.

	Pre WBVE 2		Pos WBVE 2	
BP (mmHg)	Systolic	Diastolic	Systolic	Diastolic
	107 ± 14.18	57 ± 6.55	118 ± 14.01	66 ± 17.03
HR (bpm)	75 ± 2.30		67 ± 14.00	
RR (ipm)	15 ± 1.15		15 ± 0.00	

WBVE 2 = the second intervention of whole-body vibration exercise; BP = blood pressure; HR = hearth rate; RR = respiratory rate; bpm = beats per minute; ipm = incursions per minute.

**Table 3 children-10-00213-t003:** WBVE results on tone assessed through muscle palpation.

	Pre WBVE 1		Post WBVE 1		Post WBVE 2	
	R	L	R	L	R	L
Deltoid	Hypo	Hypo	Hypo	Hypo	Hypo	Hypo
Biceps	Hypo	Hypo	Hypo	Hypo	Hypo	Hypo
Triceps	Hypo	Hypo	Hypo	Hypo	Hypo	Hypo
Breastplate	N	N	N	N	Hypo	Hypo
Abdominal	N	N	N	N	Hypo	Hypo
Abdominal Obliques	N	N	N	N	N	N
Quadriceps	N	N	N	N	N	N
Hamstrings	Hypo	Hypo	N	N	Hypo	Hypo
Triceps surae	N	Hypo	N	Hypo	Hypo	Hypo

WBVE 1 = whole-body vibration exercise sitting on the ancillary chair; WBVE 2 = whole-body vibration exercise sitting directly on the base of the platform; R = right; L = left; Hypo= hypotonia; N = normal.

**Table 4 children-10-00213-t004:** WBVE results in tone assessed through passive joint movement.

	Pre WBVE 1		Post WBVE 1		Post WBVE 2	
	R	L	R	L	R	L
Shoulder flexion	Hypo	Hypo	Hypo	Hypo	Hypo	Hypo
Shoulder extension	Hypo	Hypo	Hypo	Hypo	Hypo	Hypo
Elbow flexion	N	N	N	N	N	Hypo
Elbow extension	N	N	N	N	Hypo	Hypo
Wrist flexion	N	N	N	N	N	N
Wrist extension	Hypo	Hypo	Hypo	Hypo	Hypo	Hypo
Hip flexion	Hypo	Hypo	N	N	Hypo	Hypo
Hip abduction	Hypo	Hypo	Hypo	Hypo	Hypo	Hypo
Knee flexion	Hypo	Hypo	Hypo	Hypo	Hypo	Hypo
Ankle dorsiflexion	Hypo	Hypo	Hypo	Hypo	N	Hypo
Ankle plantar flexion	Hypo	Hypo	Hypo	Hypo	N	Hypo

WBVE 1 = whole-body vibration exercise sitting on the ancillary chair; WBVE 2 = whole-body vibration exercise sitting directly on the base of the platform; R = right; L = left; Hypo = hypotonia; N = normal.

**Table 5 children-10-00213-t005:** Body composition parameters on the WBVE protocol.

	Pre WBVE 1	Post WBVE 2
Height (cm)	118.5	119
Body mass (kg)	26	25
Lean mass (kg)	18.1	18.4
Fat-free mass (kg)	19.1	19.4
Bone mass (kg)	1.02	1.03
Total body water (kg)	14.2	14.4
Proteins (kg)	3.7	3.8
Minerals (kg)	1.23	1.23
Body fat (kg)	6.9	5.6
Muscle mass (kg)	9.2	9.4
Body mass index (kg/m^2^)	18.5	17.6
Body fat (%)	26.5	22.2
Waist-hip ratio (cm)	0.76	0.76

WBVE 1 = whole-body vibration exercise sitting on the ancillary chair; WBVE 2 = whole-body vibration exercise sitting directly on the base of the platform.

**Table 6 children-10-00213-t006:** WBVE results in DSDC.

	Pre WBVE 1	Pos WBVE 2
TS	45	43
SIMD	15	14
SBD	5	5
AD	6	3
SWTD	6	10
EDS	11	9
SH	2	2

WBVE 1 = whole-body vibration exercise sitting on the ancillary chair; WBVE 2 = whole-body vibration exercise sitting directly on the base of the platform; TS = total score; SIMD = sleep initiation and maintenance disorder; SBD = sleep breathing disorders; AD = arousal disorders; SWTD = sleep-wake transition disorders; EDS = excessive daytime sleepiness; SH = sleep hyperhidrosis.

## Data Availability

The data supporting reported results are available from the corresponding author on reasonable request.

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
