# Peer review of "Effects of the Whole-Body Vibration Exercise on Sleep Disorders, Body Temperature, Body Composition, Tone, and Clinical Parameters in a Child with Down Syndrome Who Underwent Total Atrioventricular Septal Defect Surgery: A Case-Report"

_children, 2023, doi:10.3390/children10020213_

Round 1
Reviewer 1 Report
I would like to thank the authors for dealing with such an interesting topic in such a special population. The truth is that there is a lack of research output on people with Down syndrome at this age.
A few comments that should be added:
The introduction should be enriched with research that studies the same topic in this population or a similar one (eg different genetic syndromes, mental retardation, etc.). At the end the purpose should be added clearly.
The description of the cognitive profile of people with Down syndrome needs to be made a little longer. As an example, I quote the work of Chapman, Katsarou, Andreou, Abbeduto
In the analysis of the results it is good to make a comparison between the person with Down syndrome and a person of typical development to show the difference between these two. Otherwise it seems like a simple recitation of information that does your work a disservice.
Overall, a good addition to research
Author Response
Effects of whole-body vibration exercise on sleep disorders, body temperature, body composition, tone and clinical parameters in a child with Down syndrome underwent total atrioventricular septal defect surgery: a case-report
Author's Reply to the Review Report
(Reviewer 1)
|
- Is the research design appropriate? Can be improved We appreciate your contributions and inform you that all suggested adjustments have been considered.
- The introduction should be enriched with research that studies the same topic in this population or a similar one (eg different genetic syndromes, mental retardation, etc.). At the end the purpose should be added clearly |
( ) |
(x) |
The Introduction has been improved, including suggested information (line 66-94),
- The description of the cognitive profile of people with Down syndrome needs to be made a little longer. As an example, I quote the work of Chapman, Katsarou, Andreou, Abbeduto
We wrote about this topic in the Limitations ( line 364-368) of case reporting. We inform you that cognitive assessment was not considered as a criterion in the protocol developed. Although we understand its importance, we do not have researchers qualified to perform this evaluation in our research group.
- In the analysis of the results it is good to make a comparison between the person with Down syndrome and a person of typical development to show the difference between these two. Otherwise it seems like a simple recitation of information that does your work a disservice.
There are no studies in the literature that present these results indicated in the case report, specifically found after the proposed intervention involving typical children. Our case report presents an innovative proposal, considering in addition to the diagnosis of DS, the surgical condition for correction of the ventricular septum, sleep disorders, changes in body temperature and differentiated tone. So a comparison at this point between different groups still seems early.
(Reviewer 2)
- I just recommend that this manuscript should be edited by an English professional editor for more readable. There are several grammatical errors.
We did it.
(Reviewer 3)
- Descriptive statistics (a means of describing features of a data set by generating summaries about data samples) might be provided in the manuscript, such as change BP(mmHg) from baseline by two interventions of WBVE, WBVE 1, ((|108-112|)/108)*100=3.70%; WBVE 2 ((|107-118|)/107)*100=10.28% and their implications (what are the clinical implications of this study(descriptive statistics)?)
- We included in the text a brief descriptive analysis, which corroborates the content that had already been presented in the discussion (line 177-180).

Reviewer 2 Report
This study examined to effects of whole-body vibration exercise on sleep disorders, body temperature, body composition, tone and clinical parameters in a child with Down syndrome underwent total atrioventricular septal defect surgery: a case-report. The manuscript has been well designed and written. It presents good data on this very important topic. I believe that this is very important issue in field of exercise science section.
I just recommend that this manuscript should be edited by an English professional editor for more readable. There are several grammatical errors.
Author Response

(The authors gave the same response as above.)

Reviewer 3 Report
The authors present a case report study (A 10-year-old, down syndrome girl) that aims to show the effects of whole body vibration exercise (WBVE) on sleep disturbances, body temperature, body composition, tone, and clinical parameters in a child with DS with corrected total atrioventricular septal defect (AVSD).
Comments:
2.4-. WBVE Intervention protocol,
The child performed two interventions of WBVE on a side alternating vibrating platform.
Descriptive statistics (a means of describing features of a data set by generating summaries about data samples) might be provided in the manuscript, such as change BP(mmHg) from baseline by two interventions of WBVE, WBVE 1, ((|108-112|)/108)*100=3.70%; WBVE 2 ((|107-118|)/107)*100=10.28% and their implications (what are the clinical implications of this study(descriptive statistics)?)
Author Response

(The authors gave the same response as above.)
